# Role of Senescence and Aging in SARS-CoV-2 Infection and COVID-19 Disease

**DOI:** 10.3390/cells10123367

**Published:** 2021-11-30

**Authors:** Seodhna M. Lynch, Guangran Guo, David S. Gibson, Anthony J. Bjourson, Taranjit Singh Rai

**Affiliations:** Northern Ireland Centre for Stratified Medicine, School of Biomedical Sciences, Ulster University, C-TRIC Building, Altnagelvin Area Hospital, Glenshane Road, Derry BT47 6SB, UK; S.Lynch1@ulster.ac.uk (S.M.L.); Guo-G1@ulster.ac.uk (G.G.); d.gibson@ulster.ac.uk (D.S.G.); aj.bjourson@ulster.ac.uk (A.J.B.)

**Keywords:** senescence, aging, COVID-19, SARS-CoV-2, immunosenescence, sendotypes, severity, mortality, vaccination, senolytics

## Abstract

Coronavirus disease 2019 (COVID-19), caused by severe acute respiratory syndrome coronavirus 2 (SARS-CoV-2), has resulted in a global pandemic associated with substantial morbidity and mortality worldwide, with particular risk for severe disease and mortality in the elderly population. SARS-CoV-2 infection is driven by a pathological hyperinflammatory response which results in a dysregulated immune response. Current advancements in aging research indicates that aging pathways have fundamental roles in dictating healthspan in addition to lifespan. Our review discusses the aging immune system and highlights that senescence and aging together, play a central role in COVID-19 pathogenesis. In our review, we primarily focus on the immune system response to SARS-CoV-2 infection, the interconnection between severe COVID-19, immunosenescence, aging, vaccination, and the emerging problem of Long-COVID. We hope to highlight the importance of identifying specific senescent endotypes (or “sendotypes”), which can used as determinants of COVID-19 severity and mortality. Indeed, identified sendotypes could be therapeutically exploited for therapeutic intervention. We highlight that senolytics, which eliminate senescent cells, can target aging-associated pathways and therefore are proving attractive as potential therapeutic options to alleviate symptoms, prevent severe infection, and reduce mortality burden in COVID-19 and thus ultimately enhance healthspan.

## 1. Introduction

In the 1960s, Leonard Hayflick, reported the limited proliferative capacity of cultured human diploid cells, where cells ceased to proliferate after serial passage in vitro. This was the first introduction of the permanent state of cell cycle arrest which is referred to as “cellular senescence” [1,2]. Hayflick is now respected as one of the most prominent biologists in history, although his work and research discoveries were controversial at the time as they contested Alexis Carrel’s earlier theory that cells are immortal [3]. Hayflick completely abolished this dogma which was believed for 60 years and as such give rise to a phenomenon known as the “Hayflick Limit”. The naming of this concept as the Hayflick Limit was designated in 1974 by Frank Macfarlane Burnet after Leonard Hayflick [4], and is a concept used worldwide to help explain and understand the mechanisms behind cellular aging. This concept states that human cell populations will only replicate and divide for a limited number of times before cell division completely stops and cells then enter programmed cell death or apoptosis [2]. The concept of the Hayflick Limit underpinned pioneering work by Elizabeth Blackburn, Jack Szostak and Carol Greider, who identified and discovered the impacts and effects of shortening of repetitive DNA sequences, termed telomeres, on the chromosome ends. This work facilitated scientists’ study to investigate cellular aging from embryonic development to death [5,6]. Elizabeth Blackburn, Jack Szostak and Carol Greider received the Nobel Prize in Physiology or Medicine in 2009 for their findings on telomeres and the enzyme telomerase both of which are associated with the Hayflick Limit [7]. The observation of this type of cellular senescence, first introduced by Hayflick, is also referred to as “replicative senescence” and is confirmed to be linked with progressive telomere shortening every time the cell divides [8,9,10,11].

Senescence is a very exciting, fast-moving field of research. Life expectancy worldwide is increasing, in parallel with this comes the significantly increased burden of public health and economic challenges due to increased incidence of multiple chronic diseases and disorders within the aging population, despite advances in science and medicine. Aging and the connection with senescence is acknowledged as one of the core risk factors for many of these chronic and life-threatening conditions such as cancer and cardiovascular disease. This is possibly true also for coronavirus disease 2019 (COVID-19) where it is apparent that the older population are at increased susceptibility to infection and more severe infection. Therefore, many scientists across the world are frantically racing to investigate the field thoroughly to prevent healthcare systems becoming overwhelmed.

Aging in all tissues is linked with elevated levels of cellular senescence, which is triggered by various compounding extrinsic and intrinsic factors. This stress-induced response process means cells permanently stop dividing and exit from the cell cycle and acquire a pro-inflammatory senescence-associated secretory phenotype (SASP) [12]. The acquisition of this phenotype and consequent phenotypic alterations has broad implications in health and disease [13]. As with normal cells, cancer cells can be modified to become senescent. Indeed, comprehensive understanding of this dynamic and complex process can be exploited as a potential therapeutic intervention to induce senescence in cancer cells and/or eliminate these senescent cells via a new class of agents known as “senolytics” and hence cease progression and metastasis of cancer. Furthermore, senescent cells profoundly affect tissue homeostasis, interfere with organ function, affect other cells in their environment and a similar process occurs in cancer tissues. Consequently, new therapeutics with excellent potential can be developed to induce cancer cell senescence, particularly, in resistant and metastatic cancer. 

The era of senolytics is still in its infancy and senolytic agents tested to date include Dasatinib (D, an FDA-approved tyrosine kinase inhibitor), Quercetin (Q, a flavonoid present in many fruits and vegetables), Navitoclax, A1331852 and A1155463 (Bcl-2 pro-survival family inhibitors) and Fistein (F, a flavonoid). Cardiac glycosides, such as digoxin, which is currently used to treat irregular heartbeat in atrial fibrillation, have also been reported to have broad senolytic activity [14,15]. Such senolytic compounds have been reported to function by momentarily restricting the pro-survival linkages that protect senescent cells against apoptosis without adversely impeding proliferating or quiescent, differentiated cells [16,17]. Pre-clinical animal studies have indicated that senolytics have potential clinical utility as they can eradicate senescent cells thereby mitigating age and senescence related disorders and consequently enhancing healthspan and lifespan [16,17,18,19,20]. Moreover, the results of the first human clinical trial employing senolytics have been reported. From these results, it is evident that patients with idiopathic pulmonary fibrosis, a cellular senescence-driven fatal disease, displayed improved health function after treatment with combination senolytics therapy (Dasatinib + Quercetin) [21]. Preliminary data from another recent study has shown that this same combination of senolytics therapy was able to reduce senescent cells in patients with diabetes mellitus and chronic kidney disease [22]. These studies highlight the exciting promise and potential utility of senolytics for the clinical treatment of age related and senescent associated diseases including the potential translation to cancer and potentially COVID-19. 

Even though aging is a major risk factor for COVID-19, the exact role of senescence in COVID-19 is not fully understood. Further research and studies are required to fully explore disease mechanisms and elucidate biological factors which may drive immunosenescence particularly in the elderly in the context of COVID-19. Furthermore, the timely assessment of COVID-19 severity is critical so that adequate treatments can be administrated rapidly to patients. Currently, there is a lack of definitive parameters which can be utilized to identify patients who are at high risk of requiring admission to intensive care unit (ICU). Many studies are focusing on identifying such biomarkers for this use. Indeed, changes in neutrophil-to-lymphocyte ratio and urea-to-creatinine ratio have emerged to have clinical utility as stand-alone parameters in being able to identify patients with aggressive disease at an early stage [23]. Currently, there are no such senescence-aging related biomarker signatures which have been identified for the purposes of risk stratification of severity for COVID-19 patients. Early intervention and early stratification of patients via such novel biomarker signatures will inevitably improve clinical outcomes for patients. As COVID-19 research is still very much in its infancy, further studies are needed to explore, identify, and validate potential novel stratification signatures which have robust clinical utility to identify risk of severe infection versus non-severe infection and also to help recognize patients who are higher risk of requiring ICU admission versus those patients who can be treated in non-intensive care wards. In this review, we will discuss the features of cellular senescence, the involvement of senescence in cancer and the role of the telomere clock in aging. The review also discusses the potential association between senescence, aging, immunosenescence and vaccination (Figure 1) in the context of the global COVID-19 pandemic.

## 2. Molecular Features of Senescent Cells

When cells become senescent, they undergo a plethora of significant characteristic changes including permanent, irreversible cell cycle arrest, morphological, metabolic changes, epigenetic, transcriptional as well as chromatin remodeling, metabolic reprogramming, altered gene expression, altered microRNA (miRNA) expression and the adoption of a pro-inflammatory phenotype commonly known as SASP [12]. These phenotypic alterations, which can be inter-dependent, are evident across the various stages of the senescent process including senescence initiation, early senescence, and late phases of senescence. Even though these features can be inter-dependent, for simplicity, these are described separately in the following sections. Senescence is a cellular stress response triggered by molecular damage, such as that caused by replicative exhaustion, aberrant oncogene activation (oncogene-induced senescence, OIS), or treatment with chemotherapeutics. Senescence can also be triggered in terminally differentiated cells such as neurons [24,25,26], and cardiomyocytes [27,28]. There are various types of senescence ranging from physiological senescence (e.g., embryonic development, wound healing, tissue remodeling, tumour protection, brain homeostasis), replicative senescence (e.g., telomere shortening and telomere length-dependent limit of cell divisions), stress-induced premature senescence (e.g., activation of oncogenes, inhibition of tumour suppressor genes, DNA damage, reactive oxygen species (ROS), metabolic stress, epigenetic stress, nucleolar stress, telomere independent) and drug-induced senescence (e.g., Cyclin-Dependent Kinases (CDKs) inhibitors, Histone Deacetylase (HDAC) inhibitors, Protein Kinase C (PKC) activators, genotoxic drugs) [12,29]. 

The main drivers instigating senescent growth arrest are the CDK inhibitors (CDKi) encoded in the CDKN2A (p16INK4a), CDKN2B (p15INK4b) and CDKN1A (p21CIP1) loci. It is well documented that the expression levels of these drivers are known to deviate depending on the phase of senescence and are subject to changes with progression to different phases of senescence [30,31,32,33,34,35,36,37,38]. Morphological changes evident in senescent cells have also been well documented and include the cells becoming enlarged, flattened, vacuolized, accumulation of lysosomes and mitochondria, altered composition of the plasma membrane, and sometimes they may feature enlarged or multiple nuclei and cytoplasmic bridges, such changes are controlled by various different proteins [39,40,41,42,43,44,45,46,47]. Metabolic changes evident in senescent cells and which have also previously been reported include augmented glycolysis, lysosome biogenesis, mitochondrial metabolism, and autophagy. Indeed, studies have highlighted that inhibition of autophagy enables the senescence phenomenon, and decreased autophagic activity in various tissues has been reported to be associated with aging. Autophagy therefore acts as both a biomarker of aging and a popular anti-aging target [48,49,50,51,52,53,54]. 

Epigenetic modifications, chromatin rearrangements and global elevations in chromatin accessibility have been frequently reported in senescence [55,56,57,58,59], this includes the formation of senescence-associated heterochromatin foci (SAHFs) [29,59,60,61,62,63,64], senescence-associated distension of satellites (SADSs) [56,65] and retro-transposable elements [56,66], all of which are implicated in senescence. The visual observation of such SAHFs for example can be deemed somewhat valuable for identifying senescence however the conservation across cells and functional significance of these global chromatin alterations remains to be elucidated [32,67,68] therefore they cannot be used alone as a marker indicating senescence and rather must be considered with a panel of other markers.

Altered gene and microRNA (miRNA) expression can impact and modulate the senescence process by directly or indirectly targeting key senescence factors including p53, p21CIP1 and SIRT1 [69,70]. Various studies have identified proteomic and transcriptomic signatures, many of which include coding and non-coding RNAs, which are implicated in mediating and altering senescence [30,70,71,72]. For example, many studies have illustrated that miR-24 suppresses the cell cycle regulator p16INK4a in many diseases including osteoarthritis [73] and prostate cancer [74]. MiRNAs also inhibit known suppressors of senescence including members of the polycomb complexes such as Chromobox Homolog 7 (CBX7), Embryonic Ectoderm Development (EED), Enhancer of Zeste Homolog 2 (EZH2), and Suppressor of Zeste 12 (SUZ12) (miR-26b, miR-181a, miR-210, and miR-424) resulting in p16INK4a depression and induction of senescence [75]. This highlights the potential promise miRNAs may have as novel therapeutic “senomiRs” for eliminating senescent cells [76] and thereby alleviating the detrimental impairment caused by senescence in aging pathologies. 

Finally, one of the most extensively exercised biomarkers of senescence and aging cells, detected in most in vitro and in vivo settings, is senescence associated *β*eta Galactosidase (SA *β*-gal) activity which is known to be greatly enhanced in senescent cells due to the presence of elevated lysosomal substance [77,78,79]. It is important to note that the increased lysosomal content however is also a common feature of specific cell types such as macrophages, Kupffer cells, and osteoclasts [80,81]. This emphasizes the requirement to use a panel of biomarkers in combination as opposed to relying solely on SA *β*-gal activity to distinguish senescence. Ki67, is another frequently used marker to indicate cell proliferation. Senescent cells permanently exit the cell cycle therefore this marker should not be expressed in senescent cells [82]. However, contrasting findings which have shown that Ki67 may be expressed in quiescent cells [83], and also can be expressed in senescent cancer cells [84]. Again, this highlights the need for an accurate panel of biomarkers for senescence as opposed to relying on sole biomarkers. 

In summary, many biomarkers (e.g., morphological features, SA *β*-gal staining, p21CIP1, p16INK4a, heterochromatin (SAHFs & SADSs) and proliferation (Ki-67) markers) must be used in combination as part of a panel [85], in order to correctly identify and confirm senescence in cells.

### Senescence-Associated Secretory Phenotype (SASP)

Triggers, such as cytokines, activated oncogenes, ROS, and DNA damage evoke a type of senescence, called stress-induced senescence as a response mechanism with broad implications in health and disease [13]. In contrast to apoptosis, senescent cells remain viable for a longer time and exhibit SASP. SASP is a plethora of pro-inflammatory cytokines and chemokines, responsible for the complex crosstalk between senescent cells and neighboring cells [86]. The SASP is encompassed with many immune related markers including tumour necrosis factor (TNF)-α, IL-1α/β, interleukin (IL) -6, IL-8, CC-chemokine ligand (CCL)-2 and various tissue remodeling factors including TGF-β and matrix metalloproteinases (MMPs) [12,87,88,89,90]. Accordingly, the participation of cellular senescence in various pathological processes may not only be explained by reduced proliferation but also, by cell–cell interactions and the secretion of mediators that affect surrounding cells [91]. The SASP is often likened to a double-edged sword as it is dynamic in nature [30]. This is because on one hand, prolonged exposure of SASP inflammatory mediators can consequently result in the excessive enlistment of damaging immune cells which leads to chronic inflammatory diseases [92,93,94]. Conversely, temporary SASP can promote wound healing caused by acute cellular damage and can also promote immunosurveillance, particularly in the context of tumourigenesis where they recruit T cells and natural killer (NK) cells [90,95,96]. It is important to note however, the SASP is generic, context dependent and highly diverse and is mediated at various levels. This means that the SASP is limited in its capacity as an unequivocal marker of senescence as no unique form or mechanism of SASP is known to exist [30,97,98]. 

## 3. Senescence and Cancer

Currently within the western world, the life expectancy of the global population is increasing. Predictions have reported that the global proportion of people aged over 65 years is expected to increase from 18% to 26% by 2041 and the number of people aged over 85 years will double from 2% to 4% [99]. Concurrent to the increasing life expectancy, are increasing levels of age-related conditions and chronic disease and disorders. As a result, lifespan has overtaken healthspan [100,101], and there is an overwhelming burden of chronic health complications and conditions [102] present within the older population, with over half of people over 75 years experiencing at least 2 or more chronic health conditions [99]. One of these most common age-related chronic health conditions is cancer. Cancer is known to be more prevalent in the older population and estimates report that there will be significant escalations worldwide in the next 15 years and this will be driven by the aging population and increased life expectancies [103]. 

Aging and cancer are connected to each other via extensive overlap of several molecular pathways and causes implicated in aging [104], and those also involved in cancer [105]. Aging is pivotal to the causality of numerous cancers and is known to influence several aspects including response to therapy, provision of therapy, tolerance to therapy and prognosis. Therefore, one of the most fundamental factors in the aging process, referred to as senescence, links both the aging and cancer phenomena together. It is thought that senescence has a predominant tumour suppressive function by limiting aberrant or excessive cell proliferation [106], however it has been reported that the acquired SASP and resultant alterations in the cellular microenvironment of senescent cells may in fact promote and encourage tumour progression, tumour recurrence and SASP programming may indeed drive cancer-therapy induced senescence of tumour and non-tumour cells [44,107,108,109,110,111]. This next sections of this review will endeavour to link both aging and senescence in the context of a very relevant and new disease, COVID-19.

## 4. The Potential Implications and Clinical Translation of Senescence & Aging in COVID-19

COVID-19 is caused by severe acute respiratory syndrome coronavirus 2 (SARS-CoV-2), which has a fundamental role in regulating gene expression and viral pathogenicity. COVID-19 has resulted in a pandemic associated with substantial morbidity and mortality worldwide. The identification of diagnostic/prognostic biomarkers and therapeutic targets is therefore critical in directing the pandemic. The link, if it exists, between senescence, aging and COVID-19 is an extremely novel area of research, has not been extensively studied and therefore remains an open question.

Epidemiological studies to date have shown that COVID-19 has a high mortality especially in patients of advanced age and those with comorbidities such as cardiovascular disease, diabetes, high blood pressure, chronic obstructive pulmonary disease, asthma, and cancer are much more susceptible to developing severe COVID-19 disease [112,113,114,115]. Cellular senescence can be prematurely stimulated by viral infections. Some viruses can activate senescence via DNA damage [116], or cell fusion [47]. This suggests senescence may have a key role in many viral infections.

Currently, COVID-19 research using proteomics and transcriptomic analyses are focusing on identifying and validating prognostic, diagnostic and prediction markers which are more robust and reliable than current inflammatory markers. Studies are also working to establish if the presence of specific biomarkers make a patient more susceptible to a more severe infection. The identification of such biomarkers could help predict the severity of infection expected and therefore could help determine in advance the level of medical intervention that would be much more beneficial and could be administered much faster to the patient. For example, a recent study has reported that long pentraxin 3 (PTX3) is an independent strong prognostic indicator for predicting mortality in COVID-19 and is a superior biomarker compared to conventional inflammatory markers [117]. Conversely, another study disputed this finding and has suggested that PTX3 is not any more valuable than other markers and that there are more important markers of inflammatory pathways [118]. Another study has reported that Growth Differentiation Factor 15 (GDF-15) is increased in patients who are hospitalized with COVID-19 and higher levels are associated with a worse outcome. Again, this biomarker was reported to be superior to routinely used cardiovascular and inflammatory biomarkers [119]. Other studies have also identified several differentially expressed proteins which are specific to COVID-19 severity [120,121,122,123]. As discussed previously, telomere shortening is associated with senescence. A recent study has reinforced the idea that senescence is linked to COVID-19, by establishing that patients who possessed shorter telomeres have an elevated risk of developing severe COVID-19 pathologies [124].

It is therefore likely that a combination of biomarkers will be required as opposed to individual biomarkers for accurately predicting COVID-19 clinical outcomes. Senescence markers could be combined with patient phenotypes to identify patients that might develop severe COVID-19 or predict those who will go on to have post COVID-19 syndrome, referred to as Long COVID. They may also help understand the immune response to both SARS-CoV-2 infection and vaccination. Long COVID is defined as having signs and symptoms post COVID-19 infection which continue for greater than 12 weeks and cannot be explained by alternative diagnoses. It is recognized as an evolving problem with substantial potential to impact healthcare and economics on a worldwide and longer-term basis. Patients with Long COVID have been reported to present with ongoing symptom burden such as persistent breathlessness, cough and fatigue, elevated blood biomarkers such as d-dimer and C-reactive protein (CRP) levels, have changes in lung function and deteriorating chest imaging which could lead to lung fibrosis [125,126,127,128,129]. The reasons as to why some people develop Long COVID is poorly understood, and further studies are mandated to identify drivers which predispose individuals to developing this debilitating condition. As postulated throughout this review, we propose that senescence-aging-COVID-19 are clearly linked even though the relationship and mechanisms are not yet fully apparent. We also propose that it may be possible that the development of Long COVID is also implicated within this loop; studies will be required to further explore this likelihood. The identification of hallmarks of potential senescent endotypes (or “sendotypes”) could be therapeutically exploited for drug discovery to alleviate COVID-19 symptoms, both during and post SARS-CoV-2 infection (Figure 2). Machine learning techniques could be applied in conjunction with the Horvath Clock and clinical data to identify novel secretory senescence signatures to yield sendotypes that may accurately differentiate severe COVID-19 patients from patients with mild to no symptoms. The next sections of this review will focus primarily on the associations between aging, senescence, immunosenescence and vaccination in COVID-19.

### 4.1. Aging and COVID-19

The global pandemic of COVID-19 was declared on 11 March 2020 by the World Health Organization (WHO). As per the WHO statistics, as of 17 September 2021, there have been 226,844,344 confirmed cases of COVID-19 including 4,666,334 deaths [130]. COVID-19 can present as asymptomatic or symptomatic, with the main symptoms including fever (high temperature: 38 °C or above), a new cough, shortness of breath, fatigue, loss or change to your sense of smell or taste and gastrointestinal symptoms. More severe cases of SARS-CoV-2 infection may manifest into viral pneumonia-induced acute respiratory distress syndrome (ARDS) and are associated with mortality and age [112,131]. Most cases of SARS-CoV-2 infection present as mild to moderate illness and some people do not have any clinical manifestations at all from SARS-CoV-2 infection [132,133]. Such individuals who present as asymptomatic are therefore deemed as a major sources of virus spread [132,134]. A study reported in New York illustrated that out of 141,495 patients tested, 33% of the patients tested positive for COVID-19, many of these positive cases were asymptomatic patients [135]. Thus, the extent of asymptomatic infection and asymptomatic spread is potentially a major confounding factor in controlling the global pandemic. Indeed, the silent spread of COVID-19 worldwide will ultimately increase transmission amongst all individuals and especially to the more elderly populations who are already prone to a higher risk of more severe infections and complications [136,137]. The development and implementation of various public health measures including contact tracing and prediction/forecasting models may help to address this alongside artificial intelligence and machine learning technologies [138,139,140].

COVID-19 is recognized as having a heterogeneous nature and therefore a diverse plethora of host factors are fundamental in determining the severity and progression of the disease in which a patient may acquire [133]. Some of the major risk factors associated with acquiring severe SARS-CoV-2 infection include age, male gender, smoking, chronic obstructive pulmonary disease (COPD), obesity, asthma and underlying comorbidities including chronic health conditions such as hypertension, diabetes, and cancer [115,141,142,143]. Indeed, significant evidence from across the world strongly advocates that age is the most important determining risk factor for severe COVID-19 disease [144], and its adverse health outcomes including hospitalization, moderate or severe pneumonia, severe ARDS, cardiovascular complications, kidney injury, stroke, ICU admission and death [145,146,147,148]. The following paragraphs presents the data from around the world which support the hypothesis that COVID-19 is an emergent disease of aging.

China was the first country to report SARS-CoV-2 infection amongst its population. Early data from China indicated that the case fatality ratio (CFR) of COVID-19 increases with age. It was reported that the CFR for patients aged 40 years or less was lower than 0.4%, the CFR increased to 8% for patients in their 70’s and further increased to a staggering 14.8% in patients aged 80 years and above [141,149]. Italy also reported the same profound effect of aging in its COVID-19 CFR data where the CFR was lower than 0.4% for patients aged 40 years or less and increased to a CFR of 12.8% for patients in their 70’s and further increased to a CFR of 20.2% for patients in their 80 s and above [131]. Moreover, the overall CFR in Italy was much higher than China, 7.2% vs 2.3%, respectively, which may be explained by the fact that Italy has one of the highest proportions of elderly adults in the world. Similar findings were also reported confirming this strong link between COVID-19 CFR and aging across various different countries where the CFR for COVID-19 was shown to increase exponentially with age [150]. In a separate retrospective study of 1591 patients in Italy, it was reported that the median age of patients critically ill with COVID-19 was 63 years old [136].The COVID-19 data reported by the US Center for Disease Control and Prevention (CDC) on 15 September 2021 also demonstrates that the older age group of 65 years and above accounted for 77.7% COVID-19 deaths compared to the lower age group of 45 years or less which accounted for only 3.3% COVID-19 deaths [151], again highlighting age as the major determinant factor in COVID-19 mortality.

Indeed, patient age is reported to contribute to the risk of COVID-19 incidence and severity [152]. Nursing homes (NHs) globally have been detrimentally affected with COVID-19 outbreaks and as a result, these settings are associated with increased mortality linked to COVID-19. This is mainly due to the fact that patients within NHs commonly are usually older in age, have comorbidities and tend to be frailer, thus highlighting the importance of evaluating frailty status [via the Multidimensional Prognostic Index (MPI)], to help stratify prognosis for COVID-19 patients [153]. Several studies have reported that SARS-CoV-2 infection is associated with a higher risk of mortality than those patients not affected by COVID-19 within NHs [153]. The most striking evidence is the data reported on COVID-19 cases and death in NHs across the US. This comprehensive analysis updated by The New York Times on 01 June 2021 indicated that there are up to 1.5million NH residents in the US, approximately 0.5% of the total US population. Stark figures have demonstrated that 4% of the confirmed COVID-19 cases in the US was found to be amongst these vulnerable and frail elderly patients. Sadly, they accounted for 31% of the COVID-19 fatalities in the US [154]. This is similar to other countries where NHs accounted for 41% COVID-19 deaths (based on 22 countries) even though they house only a small proportion of the total population [155]. Furthermore, it is important to establish the specific characteristics which make certain NH residents more susceptible to mortality from SARS-CoV-2 infection. One study reviewed 5,256 US NH residents who had acquired SARS-CoV-2 infection and found that increased age along with other features including male sex, impaired cognitive and physical function were independent risk factors for 30-day mortality [156]. Other studies also report similar findings highlighting specific characteristics and prognostic factors for mortality in COVID-19 positive elderly patients, these studies highlight the need for early diagnosis and individualized therapeutic management for elderly patients considering their medical history and polypharmacotherapy [144,153].

Overall, taken together, the current epidemiological data indicates that COVID-19 patients aged >80 years illustrate a higher risk of mortality compared with younger patients [115,131,144]. As the evidence demonstrates, age is considered to be one of the single biggest factors in COVID-19 with underlying comorbidities and chronic health conditions found in higher abundance in the aged. Therefore, this predisposes them to a higher risk of more severe COVID-19 disease and poorer clinical outcomes. Another aspect of concern is the impact of vaccination and how age is also implicated, this will be discussed later in this review. 

### 4.2. Senescence and COVID-19

As mentioned in previous sections, senescence is the phenomenon which describes the state when cells stop proliferating and enter permanent cell cycle arrest. A multitude of biomarkers must be used to correctly identify senescent cells. A major age-associated hallmark is the accumulation of senescent cells with the acquirement of the SASP. The next sections will endeavor to highlight how senescence and the process of biological age, rather than chronological age, is perceived as a potential mediator in COVID-19 pathogenesis and severity. Studies are required to identify functional mechanisms which link SARS-CoV-2 infection, COVID-19, and biological aging. Addressing this may potentially provide attractive avenues for therapeutic targets to alleviate COVID-19 disease severity, this is of particular interest for the elderly who are most vulnerable and susceptible.

During aging and chronic disease, senescent cells are allowed to accumulate and are not removed via immune regulation or apoptosis processes. Aged tissues are also prone to damage and thus are more likely to enter into senescence [92,157,158,159,160]. Cumulatively, these events give rise to the continual buildup of senescent cells with age and the initial benefit is quickly replaced by detrimental functions which are primarily induced by the persistent secretion of SASP factors, malfunction of processes within tissues, atypical propagation of paracrine senescence and chronic inflammation. Consequently, the process of senescence compromises tissue repair and regeneration and ultimately contributes towards the aging process. As mentioned already, it has been reported that the eradication of senescent cells via senotherapies, in aged or chronic diseases, can alleviate the commencement and advancement of age-associated dysfunctions and pathology and ultimately help to extend health span [21,22,161,162,163,164]. Studies have shown that viral infection in cells can induce cellular senescence [47,116]. The initiation of senescence can be mediated directly or indirectly by elevating IFNs secretion from viral infected cells [165], and via the discharge of damage-associated molecular patterns (DAMPs) from cells enduring inflammatory cell death [166]. Indeed, extended exposure to IFN-γ and IL-6 was demonstrated to activate senescence within the surrounding environment via SASP factors [167,168]. 

One of the principal elements of SARS-CoV-2 infection is the consequent reaction of the patient’s immune response to the virus and the subsequent multi-organ inflammatory response. Aging is associated with weak and less effective immune responses; this phenotype is defined as “immunosenescence”. The acquirement of this phenotype in the elderly predisposes them to complications instigated by viral infections [169]. A significant correlation between age and SARS-CoV-2 viral load has been demonstrated [170,171]. Viral infection via SARS-CoV-2 triggers a dysregulated reaction referred to as the “cytokine storm” during the early stage of infection. During the cytokine storm, an array of inflammatory cytokines and chemokines such as IL-6, IL-8, IL-12, IL-1β, CXCL-10, CCL-2, IFN-γ, and TNF-α are emitted. Several of these inflammatory molecules have the capacity to instigate “paracrine” senescence via a persistent cytokine signaling [112,167,168,172,173]. Infection via the SARS-CoV-2 virus is thought to instigate inflammatory cell death otherwise known as pyroptosis [174]. Even though, an immune response is normal to curtail an infection, evidence is showing that excessive levels of such cytokines and inflammatory regulators is correlated with poorer outcomes. For example, elevated levels of various markers including serum ferritin, prothrombin time, lactate dehydrogenase, and IL-6 have been linked with mortality in COVID-19 patients [115]. Findings have also shown that increased levels of cytokines have been reported in COVID-19 patients compared to healthy patients. Deeper analyses also revealed that specific concentrations of some markers including IL10, GCSF, IP10, MCP1, MIP1A, and TNFα were elevated in ICU COVID-19 patients compared to non-ICU patients [114]. This massive cytokine storm apparent in infection, in conjunction with altered anti-inflammatory functions, may contribute to an imbalance in the coagulation axis consequently impacting on severity and leading to fatal clinical outcomes [175,176,177,178,179]. These manifestations are phenotypical of immunosenescence, and these will be discussed further in the next sections of this review. 

In brief, it is expected that patients of increased age have a higher probability of accumulating high levels of cellular senescence. The rationale behind this is that the proportion of senescent cells is already increased prior to the time of SARS-CoV-2 infection thus may accelerate paracrine senescence events. Older cells are also more prone to becoming senescence due to the reduced ability to restore impairments. Finally, an aged immune system is less competent in its ability to eradicate senescence cells, this is partially due to a deterioration in immune functionality. This may explain why older patients are more susceptible to SARS-CoV-2 infection and at a higher risk of more severe COVID-19 disease and mortality. It is thought that inhibition of SASP or reducing senescent lung cells via senolytics may be more beneficial in elderly people with COVID-19. This is a hot topic and new research findings may have a significant translational impact on patients by identifying novel senescent biomarker signatures which have the potential to change the way we predict COVID-19 severity. This may provide exciting avenues for therapeutic intervention by targeting senescence-associated mechanisms via senotherapies in COVID-19 patients and may also serve to help enhance vaccination efficacies. 

### 4.3. Immune Dysregulation during COVID-19

SARS-CoV-2 infection activates both innate and adaptive immune responses. In at risk individuals, with diminished anti-viral defenses, an excessive immune response can develop which is associated with aberrant inflammation and increased disease severity [179,180]. This immunopathological response can exacerbate comorbid conditions, spark disproportionate clotting and cardiovascular complications, and cause persistent symptoms such as fatigue and depression. In severe disease subgroups immune system dysregulation is evident; elevated plasma cytokine levels are accompanied by reduced T lymphocyte numbers (especially CD4+ and CD8+ T cells), accumulation of neutrophils, fever, platelet deficiency, elevated ferritin, and elevation of other inflammatory and clotting factors [112,177,181]. Our knowledge of the discrete mechanisms and signatures of the pathological immune response and its impact upon the development and persistence of protective immunity is developing at pace with longitudinal, immunological, omics and systems biology approaches.

Immunological responses detected post-SARS-CoV-2 infection are highly variable, though appear to be associated with severity and influenced by age and comorbidities. Patients with severe disease tend to have higher antibody titres than those with mild disease, possibly due to higher viral load [182]. Older age groups where the immune system is often functionally impaired also tend to have elevated antibody titres, though the functional quality and durability of protection is unknown [182,183,184,185]. Also male, older, and obese patient groups tend to have higher background inflammation and lower T cell subsets, these are already known to be depleted in severe COVID-19, the presence of these characteristics may therefore increase susceptibility to SARS-CoV-2 and predisposes individuals to more severe disease [186]. Furthermore, the level of T cell response appears to be important, as poor T cell immunity correlates with severity and potentially short-lived protective immunity [187]. 

Transcriptome wide analyses of blood from severe COVID-19 patients reveals a series of activated immune system genes including those associated with neutrophils, secretory granules, neutrophil extracellular traps (NETS) along with clonal expansion of adaptive immune response and diminished lymphocyte gene expression [188]. The following sections provide an overview of the exuberant innate and poor adaptive immune responses and subsequent cytokine storm and tissue damage sequalae observed in severe cases of COVID-19.

#### 4.3.1. Innate and Adaptive Immune Responses in the Pathogenesis of COVID-19

Initial innate system Type I and III interferon responses are suppressed or disrupted which results in early IL-6, IL-10 and IL-1β enhanced inflammation [189]. It has been postulated that SARS-CoV-2 along with other coronavirus’ show genome patterns (significant under-representation of CpG dinucleotide pairs) associated with diminished viral recognition and dysregulated immune response [190]. The ensuing inflammation is propagated by aberrant monocytes, macrophages, and NK cells in their attempt to send viral pathogen-associated molecular patterns (PAMPs) and DAMPs into affected tissues. This results in neutrophil-driven pathology including fibrosis that causes ARDS. Activated leukocytes and neutrophil extracellular traps (NETs) also promote abnormal clotting which accumulates in lung and kidney tissues of patients with severe COVID-19 [191,192]. Nucleocapsid and spike proteins of SARS-CoV-2 are thought to play a central role in driving NET formation [193]. Treatments that target inflammation and coagulation may therefore reduce COVID-19 mortality.

CD4+ and CD8+ are an important part of the antiviral adaptive immune response to SARS-CoV-2 infection. The quality and breadth of CD4+ and CD8+ T cell response plays a key role in COVID-19 resolution and modulation of disease severity. B and T cell depletion are early indicators of severe disease development and along with markers of liver damage, act as predictors of mortality in severe COVID-19 patients [194]. Regulatory T-cells on the other hand control immune system homeostasis and self-tolerance by negative regulation of effector functions, activation, and proliferation of immune cells [195]. Increased Notch 4 expression on circulating regulatory T cells (Tregs) promotes inflammation and diminishes repair of lung tissue and again predicts mortality in those severely affected by COVID-19 and in convalescent patients who exhibit reduced levels post recovery [196].

In recovered COVID-19 patients and in terms of resultant protective immunity, an integrated study of SARS-CoV-2 immune memory in a longitudinal cohort over 8 months exhibited distinct kinetics [184]. Immunoglobulin G (IgG) to the spike protein is relatively stable over 6 months. Spike-specific memory B cells are more abundant at 6 months than at 1 month after symptom onset. SARS-CoV-2 specific CD4+ T cells and CD8+ T cells decline with a half-life of 3 to 5 months. Investigations of T cell reactivity across the entire SARS-CoV-2 proteome reveal distinct patterns of immunodominant S, N, and M protein regions for CD4+ T and CD8+ T cells [197]. Tarke et al. also found negligible impact of mutations found in SARS-CoV-2 variants upon CD4+ and CD8+ responses in those recovered from COVID-19 or mRNA vaccination [198]. Persistent symptoms also do not adversely impact SARS-CoV-2 specific T cell-based immunity [199].

#### 4.3.2. The Cytokine Storm and its Association with COVID-19 

There is continued debate around the role of the so-called “cytokine storm” in COVID-19 pathophysiology and severity [200]. Albeit there is irrefutable evidence of increased systemic levels of pro-inflammatory cytokines in subsets of patients with COVID-19, elevated cytokine response is more consistent in other acute conditions such as sepsis, trauma, and cardiac arrest. Imbalanced cytokine responses for example, rather than magnitude of cytokine storm may contribute to cardiac dysfunction in juvenile multisystem inflammatory syndrome (MIS) [201]. A tissue wide systems immunology-based study reveals that the cytokine storm triggered by SARS-CoV-2 infection may result from dysregulated cytokine production by inflamed pulmonary, heart, liver, and kidney tissues [202]. Further studies are warranted to further explore the association between COVID-19, immune dysregulation, and cytokine storm. 

### 4.4. Immunosenescence and COVID-19 Severity and Mortality

Immunosenescence is a distinctive feature of aging and is associated with defective immune responses and thus less effective anti-viral responses. As individuals get older, there is aberrant interruption of both the innate and adaptive immune responses alongside chronic inflammation due to recurrent production of inflammatory mediators and cytokines, also referred to as “inflammaging”, thereby predisposing the elderly to complications during viral infections [169,203,204]. Immunosenescence has pleiotropic effects on the immune system and changes observed can be attributed to numerous intrinsic and extrinsic factors. Immunosenescent phenotypic manifestations include dysregulation of innate immunity and cytokine production, loss of naïve T cells, accumulation of terminally differentiated memory T and B cells, deterioration in the function of T and B cell function and decreased lymphocyte proliferation [205,206,207]. 

There are limited studies on the role of cellular senescence and specifically immunosenescence in COVID-19. Recent findings have however shown that there is the presence of an immunosenescent phenotype which appears to be underpinned by elevated neutrophils-to-lymphocytes ratio (NLR) [208], and high IL-6 production [209], and is correlated to disease severity [210]. For example, a recent study has reported that a Th2/Th1 cytokine imbalance is correlated with a higher risk of mortality from COVID-19 [211]. Findings from another study have reported that biological age measure [on 42 biomarkers] captured using PhenoAge, as opposed to chronological age, is responsible for pushing the trends and correlations we are seeing with COVID-19 severity and age. This study reported that accelerated aging 10-14 years prior to the COVID-19 pandemic was linked with test positivity and mortality from SARS-CoV-2 infection [212], thus highlighting that COVID-19 severity can be predicted by earlier evidence of accelerated aging. It is also thought that polymorphisms of the STING (stimulator of IFN genes, encoded by TMEM173) pathways could be implicated in COVID-19 pathogenesis [213]. Findings have shown that STING aberrations are correlated with risks of aging-related disorders. Specifically, the STING p.R293Q offers protection both from inflammaging and obesity-associated cardiovascular disease in advanced age individuals [214]. This may suggest that overload of the STING pathway may contribute towards COVID-19 severity in the elderly and also may explain complications associated with COVID-19 including myocardial infarction. Data however is limited, and it remains poorly understood if senescent cells are friends or foes in viral infections and further studies are warranted to fully explore this relationship [215].

### 4.5. Senescence Involvement in Multimorbidity and COVID-19 

Many hospital programs and services for treatment of health conditions within patients has been detrimentally destabilized due to the COVID-19 pandemic. In particular, cancer services including elective surgeries have been severely disrupted. There is now a tsunami of patients on extremely long waiting lists. Such emergency and elective surgery programs need to be separated from the COVID-19 pandemic and need to instead run in parallel. Multimorbidity (multiple long-term conditions) is defined as two or more diseases (usually >5) occurring simultaneously in an individual (usually increasing in number with age) [216], necessitating multiple treatments simultaneously. One of the greatest challenges that healthcare services face is developing optimal treatments for such patients as well as implementing new strategies and approaches to deal with the increased demand. These challenges are further exacerbated by an aging population, due to increased life expectancy which equates to increased number of patients living with multimorbidity who are seeking treatments [99]. Medical advancements have increased the lifespan but with intensifying levels of chronic disease burden and increased treatment demands, research now needs to focus on enhancing healthspan alongside lifespan. 

COVID-19 is an emergent disease of aging and poorer clinical outcomes are evident in patients with existing comorbidities [113,145,150]. Even though older and multimorbid patients represent the highest risk group, they have been excluded from the current COVID-19 vaccine trials. This exclusion of the most vulnerable will undermine and raise trivial questions as to the applicability and relevance of clinical trial data to these highest risk patients [217]. Indeed, the identification of high-risk patients in the context of their age and multimorbidity profile can be used to define potential interventions of selective confinement or unique management [218]. In addition to age, other conditions, such as cortisol excess in COVID-19 for example, are risk factors which may expose patients to having worse clinical progression following SARS-CoV-2 infection [219]. Indeed, implementation of education strategies for high-risk patients who have such conditions, in conjunction with enhanced medical management solutions such as performing clinical triage to prioritise medical consultations and digital telehealth solutions will help with self-management strategies to prevent complications and disease transmission.

The intricate means by which the phenomenon of senescence contributes to multiple long-term conditions is not yet known. We therefore hypothesise that specific sendotypes may indeed act as differentiating factors in multimorbid diseases as well as in COVID-19 and may have potential to act as therapeutic targets for intervention. We also postulate that senescence may be involved in driving more severe COVID-19 especially in elderly patients burdened with multimorbidity. 

### 4.6. Senescence Involvement in Vaccination and the Association with COVID-19

Vaccination against COVID-19 unified scientific communities globally as they raced together to identify therapeutic and preventative solutions including the identification and development of viable vaccine candidates to stem the threat of COVID-19 worldwide. There are now several vaccines which are in use across the globe, including the Pfizer-BioNTech messenger RNA (mRNA) vaccine, BNT162b2 and the Oxford-AstraZeneca adenovirus vector vaccine, ChAdOx1-S, which were approved for emergency use [220,221] by the Medicines and Healthcare products Regulatory Agency (MHRA)As reported by the WHO, as of the 15 September 2021, a total of 5,634,533,040 vaccine doses have been administered worldwide [130]. Vaccinations are administrated to the muscle where there is an abundance of senescent cells in correlation with increased age [222,223], this potentially may be a factor which may contribute to waning vaccine efficacy over time, especially in the elderly. 

#### 4.6.1. Initial Vaccination Efficacies

The COVID-19 vaccines currently available have demonstrated very high levels of efficacy during the clinical trials and these levels of efficacy are being continuously monitored and analyzed as new SARS-CoV-2 variants of concern (VOC) and variants of interest (VOI) emerge. Seropositive SARS-CoV-2 patients are projected to have 89% protection from reinfection [224], whilst reported vaccine efficacies range from 50-95% [225]. For example, the clinical trial of the COVID-19 mRNA vaccine BNT162b2 (Pfizer-BioNTech Vaccine) observed that two doses of the vaccine can confer 95% protection against COVID-19 in patients aged 16 years and over [226]. Indeed, other data has shown that one dose of either the BNT162b2 or the ChAdOx1-S (Oxford-AstraZeneca Vaccine) provides ~60-70% protection against symptomatic COVID-19 and also offers ~80% protection against hospital admission [227]. All vaccines developed to date are continuously monitored. For instance, data from four separate clinical trials in the UK, Brazil and South Africa was pooled in an interim analysis to evaluate the ongoing efficacy of the ChAdOx1-S vaccine. It was found that the vaccine demonstrated efficacy against COVID-19 across all trials [228]. 

As elderly individuals are most at risk of developing more severe COVID-19, the safety and efficacy of vaccines in the elderly is critical to their success. Studies to date on the efficacy of the vaccines in the elderly individuals have shown promising results. The administration of one dose of either the BNT162b2 or ChAdOx1-S in elderly patients specifically was associated with a significant reduction in symptomatic COVID-19. Furthermore, one dose of either vaccine was 80% effective at preventing hospital admission, and notably, a single dose of the BNT162b2 was 85% effective at preventing mortality in patients with COVID-19 [227]. This highlights the promise of vaccination for saving lives in the older population who predominantly are affected with more severe disease. 

Clinical trials are also ongoing within some countries to investigate and establish if a patient can have a first dose from one vaccine and the second dose from a different vaccine. Preliminary data from a Spanish clinical trial, CombiVacS, of ~600 patients who received AstraZeneca as the first dose followed by Pfizer as the second dose has been reported. Results so far indicate, that the mix and match of two different vaccines was able to elicit an extremely potent immune response and antibody levels increased dramatically upon the second dose with Pfizer [229]. This strategy of heterologous prime and boost, which has been deployed for vaccines against other infectious diseases, such as Ebola [230], shows the promising potential of using a combined approach. It is questionable however whether a third booster dose would work to prolong immunity or protect against emerging variants in this context and if elderly patients would benefit more from this combination vaccination strategy. More research is required before this can be confirmed.

#### 4.6.2. Emerging Impact of Neutralization Responses to Vaccination 

As the vaccines against COVID-19 are relatively new, it is still too early to know the exact duration of the protection of the vaccine and the level of immunity these vaccines offer against new emerging variants and also against reinfection. As mentioned already, current vaccines appear to have excellent efficacy when administered using the two-dose approach and data published early in 2021 demonstrated that immunological memory in 95% of individuals was retained for ~6 months after infection [184]. However, data is starting to now accumulate illustrating the impact of neutralizing responses to vaccination and the resultant correlation with waning protection in individuals and importantly in the elderly. Indeed, multiple animal models of COVID-19 have already illustrated that protection from COVID-19 is largely mediated by the antibody immune response and neutralizing antibodies [231,232,233], therefore highlighting the fragility associated with waning immune response.

Current studies are focusing on investigating these waning immune responses. A recent study has analyzed the association between in vitro neutralization levels and the apparent protection from COVID-19 infection using data from seven different vaccines and from different patient cohorts. The study highlighted that neutralization levels are highly predictive of immune protection in COVID-19 infection and that their prediction models show that deterioration in the neutralization titre occurs over the first 250 days after vaccination, resulting in a significant drop in protection from COVID-19, albeit protection against severe infection should remain [234,235]. Other studies have also found similar findings that the primary immune responses are inevitably waning [184,236,237], and there has been a considerable increase in COVID-19 positive cases amongst the unvaccinated elderly [238,239]. One of the most recent studies has shown that the efficacy of BNT162b2 vaccine declines gradually over a period of 6 months post vaccination. Despite this decline in efficacy, the study confirms that the vaccine still exhibits a promising safety profile and still remained highly effective in preventing COVID-19 [240]. 

There is limited data available on neutralizing antibodies or vaccine efficacy in the elderly, the most vulnerable older group of patients. Increasing evidence shows that both humoral and cellular immunity are both vital in fighting against COVID-19 [241]. A group of proteins known as stress-inducible proteins (sestrins) are known to have a fundamental role in controlling T cells that possess senescent like characteristics. Previously, it was shown that the knockout of these sestrins in animals resulted in the production of elevated levels of neutralizing antibodies against influenza infection [242], whilst other studies illustrated that the knock down of sestrins was able to restore the function of T cells [243]. This portrays the key function of sestrins in bridging the link between senescence and immunity. Indeed, it is speculated that cellular senescence may promote a deviation from adaptive immunity towards innate immune responses in the elderly with a negative impact on vaccine efficacy. Furthermore, recent studies have indicated that serum neutralization and binding of the IgG/IgA immune response after the first dose of vaccination decreased in patients of increased age with the most prominent decline in patients aged 80 years and above. Moreover, it was apparent that individuals aged 80 years and above had dramatically decreased or lack neutralization potency against certain SARS-CoV-2 VOC including B.1.1.7, B.1.351 and P.1 after the first dose of vaccination. However, following two doses of vaccination, this was no longer evident and neutralization against VOC was measurable irrespective of age [244]. Studies are further required to specifically address the mechanisms behind these diminished levels of neutralization antibodies and the concomitant diminished protection in the vaccinated elderly. On a contrary perspective, it is also important to note that SARS-CoV-2 infection in vaccinated individuals was more likely to present as asymptomatic, especially in those aged 60 years and above [245], this may have implications for transmission rates and also booster vaccinations. 

#### 4.6.3. Future for Booster Vaccination Implementation

Cumulatively, these findings affirm that specific measures such as booster vaccination campaigns, in the elderly especially, to enhance immune responses may need to be implemented, particularly where SARS-CoV-2 VOC are imminent. Booster vaccination programs are already underway in some countries, including Israel. Indeed, preliminary findings have shown that a third Pfizer booster shot for those aged above 60 years was able to reduce the risk of infection by 86% and reduced the risk of severe COVID-19 by 92% [246]. Other studies also show that the third booster shot is capable of inducing vaccine effectiveness increases and restoration of protection in those individuals infected with the Delta variant [247,248]. Many other countries, including Ireland, UK and the USA are now also rolling out mass booster vaccination campaigns. Overall, it is evitable, that booster doses to restore immunity at some stage will be necessary to control the COVID-19 pandemic. However, a key question is, will the third booster dose be enough to protect the high-risk populations including the elderly, or are we looking at a fourth booster or even the implementation of annual vaccination schemes similar to that for influenza in the future.

Studies highlight the utility of neutralization and immune protection prediction models in developing vaccine approaches to control the future trajectory of the COVID-19 global pandemic. Neutralization antibody levels wane over time, though this parameter alone is not specific as a correlate of immunity and must be considered with the innate/adaptive immune responses and immunological memory. It can be speculated that senescent cells, which are found at higher abundance in older people [159,160], may be directly or indirectly influencing the immunological memory capabilities of T and/or B cell functions in the immune response to SARS-CoV-2 infection and vaccination [249]. Findings have illustrated a negative correlation between CD4/CD8 T-cell ratio and the severity of frailty in the elderly in viral infection [206,250,251,252], however, the performance of these cells in COVID-19 needs to be explored. Previous studies have confirmed that proficient B cells decrease with age whilst the number of senescent memory CD27-B cells increases in the elderly [253]. We therefore postulate that there may be an impairment of both effector memory T cells and efficient B cells during SARS-CoV-2 infection which may reflect the effectiveness of the vaccination similar to that seen in influenza-like illness where the effectiveness of vaccines is lower in the elderly compared to younger individuals [254,255,256]. This plausible speculation however remains to be elucidated.

#### 4.6.4. The Potential Impact from the Emergence of New Viral Variants

The emergence of new viral variants is a persistent concern as they may have increased transmissibility [257], and they may evade control by both vaccine induced and convalescent immune responses by having reduced sensitivity to vaccine or infection elicited antibodies [258,259]. Consequently, this may impede clinical efficacy [260], thus ultimately raising fears for the most vulnerable groups, such as the elderly, where the level and quality of immune responses may be suboptimal. According to the WHO, there are four VOC to date, these are referred to as Alpha, Beta, Gamma and Delta [261,262]. The Delta variant is currently the most dominant variant worldwide and has been identified in over 170 countries and is recognized as the most dangerous, as it is much more transmissible in nature. This VOC have been recognized to be less well neutralized by vaccine induced antibodies [263]. One study has illustrated that natural immunity appears to elicit a longer lasting and more robust protection against SARS-CoV-2 Delta variant infection, symptomatic disease and hospitalization compared to BNT162b2 vaccine induced immunity [264].

Furthermore, it has been reported that vaccine effectiveness after one dose of the vaccine (BNT162b2 or ChAdOx1-S) was markedly lower in those who subsequently became infected with Delta variant, compared to those with the Alpha variant. This highlights the importance of sustaining efforts to maximize vaccine uptake of both first and second doses especially in high-risk populations such as the elderly [265]. Furthermore, although a small study, it has been reported that the Delta variant appears to be able to partially evade neutralizing antibodies stimulated by previous infection with SARS-CoV-2 or by vaccination [266]. Moreover, it has been reported that in NH residents, many of whom are elderly, vaccine effectiveness during the widespread circulation of the Delta variant was found to be significantly decreased in patients compared to the pre-Delta variant period [267]. This again emphasizes that specific prevention measures (e.g., vaccination amongst staff, visitors, residents, booster vaccinations) to protect high risk populations such as elderly residents within NHs are absolutely critical in order to optimize protective immune responses and ultimately save lives. This raises fears for the high-risk populations albeit more studies are required to fully comprehend the role of such humoral responses in the efficacy of vaccines against circulating and new emerging variants which may have specific mutations that may enhance the virulent properties of the viral.

#### 4.6.5. Potential Role of Senescence and Aging in Curtailing the COVID-19 Pandemic

Overall, the take home message is that the combination of neutralizing antibody titres due to both age and VOC/VOI, must be considered when developing and designing strategies for booster vaccinations and the duration of protection that would be conferred by such additional booster doses and the clinical impact this would have on severe COVID-19. It is also important to consider that this is fundamental in high-risk populations, such as the elderly, who also have high levels of senescence cells. Further studies are also needed to address the transcriptional, epigenetic, and functional reprogramming, termed “trained immunity”, which occurs and drives immunological memory in those who have recovered from COVID-19 [268,269,270]. One may query, is it plausible that these high levels of senescence cells may play a key role in the immune response through involvement in some of this transcriptional, epigenetic, and functional reprogramming, is waning immunity and protection seen in the elderly especially due to senescence and is senescence partially responsible for diminishing the immunological memory of T and B cells. If senescence is involved, it could be an option for senolytics to be considered as a potential therapeutic intervention. 

## 5. Conclusions and Future Perspectives of Senescence as a Promising Biomarker and Target in COVID-19 

The effects ofSARS-CoV-2 infection and resultant severe COVID-19 symptomology is likely driven by the highly destructive pathological hyperinflammatory response which results in a derailed cytokine storm, thrombosis, vascular leakage, organ damage and aberrant antibody immune responses, ultimately leading to high morbidity and mortality in the most vulnerable. Our review highlights that senescence and aging together, potentially play a central role in COVID-19 pathogenesis. 

As alluded to already, therapeutic invention via senolytics is paving the way forward in the field of aging research. Indeed, we draw upon the potential of utilizing senolytics as viable therapeutic interventions or targets in COVID-19 to prevent severe infection in patients and specifically aged patients. Strong evidence has been reported recently in old mice where senescent cells became hyperinflammatory upon exposure to the SARS-CoV-2 virus. This resulted in enhanced SASP production of pre-existing senescent cells, increased senescence, inflammation, and high mortality rates in these aged mice. Furthermore, this study went on to show that the pharmacological removal of senescent cells via senolytic (Fisetin or Dasatinib + Quercetin) treatment, in these aged mice, was shown to significantly reduce senescence, inflammatory markers and mortality associated with COVID-19 [271]. These findings demonstrate the hypothetical rationale behind the therapeutic potential of senolytics, particularly in aged patients who have a higher burden of senescence, to interrupt the initial triggered immune cascade, cytokine storm and hyperinflammation evident in severe cases of SARS-CoV-2 infection. Indeed, Quercetin and Fisetin, which have excellent safety profiles thus serve as attractive candidates, are currently being investigated in COVID-19 trials as potential senolytic compounds for early intervention [272,273,274]. Moreover, it will be intriguing to explore if such strategies of early senolytic intervention could help alleviate the chronic post-COVID-19 syndrome, Long COVID [275]. Interestingly, findings from a recent study have shown that SARS-CoV-2 viral infection and subsequent virus induced senescence (VIS) is a pathogenic driver of dysregulated cytokine storm and tissue damage. This study found that patients infected with SARS-CoV-2 had markers of senescence present in their airway mucosa and also had increased levels of SASP factors within their serum. Excitingly, they showed that treatment with senolytics eradicated VIS cells, alleviated COVID-19 lung disease and reduced inflammation in two COVID-19 driven animal models [276]. This study highlights the potential clinical promise of senolytics as a novel treatment against COVID-19, of which could also be translated to treating other viral infections and ultimately enhance healthspan. This study along with a recently published review [277], affirms our hypothesis that the senescence-aging-COVID-19 axis along with the aging immune system has a paramount role in COVID-19 and a full understanding of the mechanistic networks underpinning this axis will be vital in the future. 

The COVID-19 pandemic has identified gaps in our current knowledge of the aged immune system and the integral involvement of specific immune cell subsets. Even though aging is considered to be one of the highest risk factors for COVID-19, the biological mechanisms behind this are not fully known. Indeed, the role of senescence and immunosenescence in COVID-19 is not fully known either. Despite the large numbers of scientific articles and preprints which are being published almost daily, the exact mechanisms behind this conceptual axis of senescence-aging-COVID-19, the phenomena of immunosenescence, novel sendotypes (senescent endotypes specifically) and immune regulation are not fully known. Understanding the biological pathways involved in this disease will ultimately lead to the identification of key genes and pathways which are linked to senescence-aging-COVID-19, thus paving the way to developing novel drug targets and biomarkers for this disease. Indeed, a recent study which combined gene co-expression network analysis with artificial intelligence approaches has identified a potential novel gene signature which may have clinical value in the pathogenesis of COVID-19 [278].

Therefore, extensive studies are required to further elucidate the mechanisms as to why some patients develop more severe COVID-19 compared to others and to identify the specific aspects of the aged immune system such as novel sendotypes (senescent endotypes) which predispose the elderly to more severe clinical outcomes including mortality. It will also be interesting to explore whether senescence is a confounding factor in waning immune responses especially in the elderly and whether intervention via senolytics could mitigate decreases evident in immune responses after vaccination. It would also be interesting to explore if senescence is a factor involved in predisposing patients to Long COVID. Finally, as many people worldwide have been infected with SARS-CoV-2 and as the pandemic is very much still in its infancy and the long-term impacts from COVID-19 are yet to be fully understood and it is likely that COVID-19 may instigate a long-lasting genomic imprint in humans and their associated physiological processes. 

## Figures and Tables

**Figure 1 cells-10-03367-f001:**
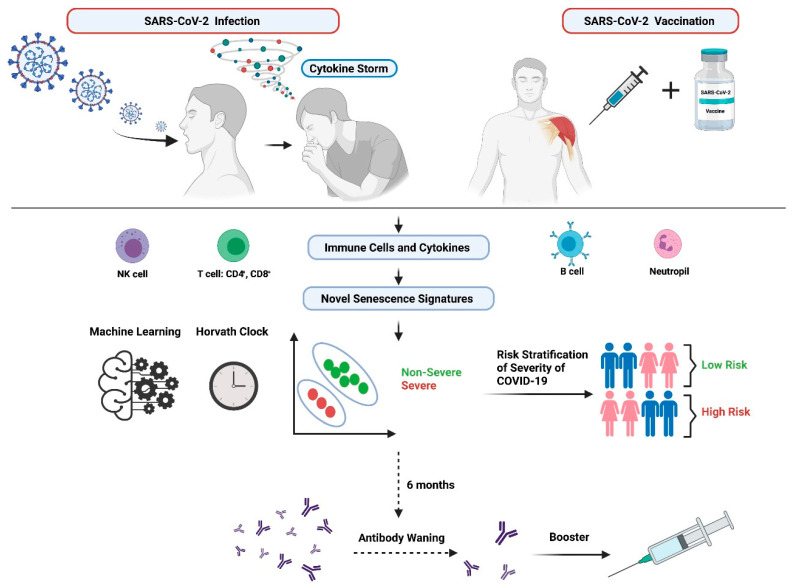
Senescence and Immune Dysregulation in COVID-19. SARS-CoV-2 infection and vaccination are both associated with major immunological alterations. These are linked to both aging and senescence. The pathological hyperinflammatory response evident in SARS-CoV-2 infection and the sub-optimal antibody immune responses following vaccination against SARS-CoV-2 may be regulated by novel senescence signatures. Identification of novel senescence signatures in combination with applied machine learning techniques and the Horvath Clock may help to identify novel senescence signatures that may accurately differentiate severe COVID-19 patients from patients with mild to no symptoms (non-severe). It may also identify novel senescence signatures which are implicated in the waning antibody responses evident following vaccination and booster vaccinations. Figure generated using Biorender.com accessed on 17 November 2021.

**Figure 2 cells-10-03367-f002:**
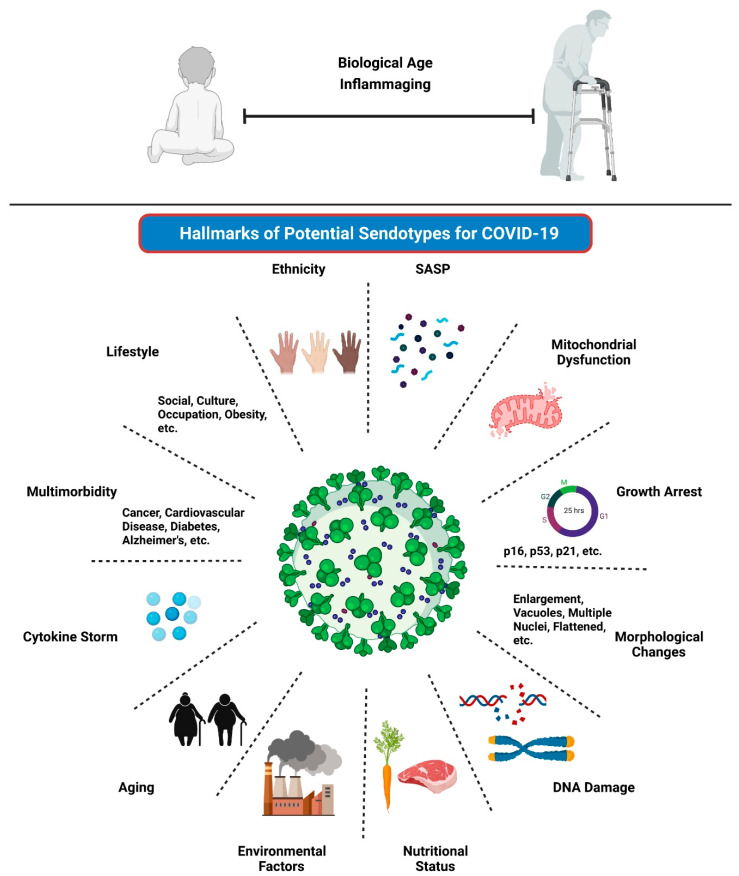
Hallmarks of Potential Sendotypes for COVID-19. The diagram outlines the hallmarks of potential sendotypes in COVID-19. The hallmarks listed can be used in combination to identify specific senescent endotypes (or “sendotypes”), which can used as determinants of COVID-19 severity and mortality. Indeed, identified sendotypes could be therapeutically exploited for therapeutic intervention via senolytics, to alleviate symptoms, prevent severe infection, and reduce morbidity and mortality in COVID-19. Future studies are required to understand the mechanistic underpinnings of each hallmark and how it would contribute towards specific sendotypes. Figure generated using Biorender.com accessed on 17 November 2021.

## Data Availability

Not Applicable.

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
