# Peer review of "Role of Senescence and Aging in SARS-CoV-2 Infection and COVID-19 Disease"

_cells, 2021, doi:10.3390/cells10123367_

Round 1
Reviewer 1 Report
The topic touched upon is undoubtedly important, promising, interesting and new. One important comment. The proposed material is redundant. There is the information that is not related to the stated topic. For a better perception the article needs significant reduction. I propose to delete sections 2.2-2.7; 3.1; to shorten 4.6.
Author Response
Reviewer 1:
Comment: The topic touched upon is undoubtedly important, promising, interesting and new. One important comment. The proposed material is redundant. There is the information that is not related to the stated topic. For a better perception the article needs significant reduction. I propose to delete sections 2.2-2.7; 3.1; to shorten 4.6.
Response: We thank the reviewer for these comments. We have deleted the sections 2.2-2.7; 3.1;
We have substantially shortened section 4.6 and have sub divided the remaining content into subsections for better flow of information

Reviewer 2 Report
Seodhna M Lynch uncovered a broad spectrum of biological and clinical aspects of senescence and aging in SARS-CoV-2 infection and COVID-19 disease.
1) The rationale of why the authors came up with this review.
2) What is the information that is not exactly available that motivated the authors to come up with this information. What are the current caveats and how do the authors highlight the current research in answering them? If not they need to address in future directions.
3) I personally miss some biological insights in the introduction: a short flash on short-time changes in neutrophil-to-lymphocyte ratio and urea-to-creatinine ratio emerged as stand-alone parameters able to identify patients with aggressive disease at an early stage. Can the author expand (refer to PMID: 33553217);
4) in the frame of this thinking, he identification of key genes and pathways in this disease may lead to finding potential drug targets and biomarkers and can boost the interest for a broad readership considering these authors' review (refer to PMID: 34441862);
5) Next, some endocrine conditions, such as cortisol excess, may be risk factors of worse clinical progression once the infection has occurred. These at-risk populations may require adequate education to avoid the SARS-CoV-2 infection and adequately manage medical therapy during the pandemic, even in emergencies. Endocrine disease management underwent a palpable restraint, especially procedures requiring obligate access to healthcare facilities for diagnostic and therapeutic purposes. Strategies of clinical triage to prioritize medical consultations, laboratory, instrumental evaluations, and digital telehealth solutions should be implemented to better deal with this probably long-term situation. Can the authors expand on that?
6) The authors need to highlight what new information the review is providing to enhance the research in progress.
Author Response
Reviewer 2:
Comment: The rationale of why the authors came up with this review.
Response: We thank the reviewer for this comment. We have now added the rationale to introduction
Comment: What is the information that is not exactly available that motivated the authors to come up with this information. What are the current caveats and how do the authors highlight the current research in answering them? If not they need to address in future directions.
Response: We thank the reviewer for this comment. We have now addressed this in future direction
Comment: personally miss some biological insights in the introduction: a short flash on short-time changes in neutrophil-to-lymphocyte ratio and urea-to-creatinine ratio emerged as stand-alone parameters able to identify patients with aggressive disease at an early stage. Can the author expand (refer to PMID: 33553217);
Response: We thank the reviewer for this comment. We have now addressed this in the introduction section and also discussed findings of PMID: 33553217 in the same section.
Comment: in the frame of this thinking, he identification of key genes and pathways in this disease may lead to finding potential drug targets and biomarkers and can boost the interest for a broad readership considering these authors' review (refer to PMID: 34441862)
Response: We thank the reviewer for this comment. We have now addressed this in the future directions section and also discussed findings of PMID: 34441862 in the same section.
Comment: Next, some endocrine conditions, such as cortisol excess, may be risk factors of worse clinical progression once the infection has occurred. These at-risk populations may require adequate education to avoid the SARS-CoV-2 infection and adequately manage medical therapy during the pandemic, even in emergencies. Endocrine disease management underwent a palpable restraint, especially procedures requiring obligate access to healthcare facilities for diagnostic and therapeutic purposes. Strategies of clinical triage to prioritize medical consultations, laboratory, instrumental evaluations, and digital telehealth solutions should be implemented to better deal with this probably long-term situation. Can the authors expand on that?
Response: We thank the reviewer for this comment. We have now addressed this in section 4.5 and also discussed findings of J Clin Med 2021, 10 in the same section.
Comment: The authors need to highlight what new information the review is providing to enhance the research in progress.
Response: We thank the reviewer for this comment. We have now addressed this in section 5.
